# Computationally Efficient Direction-of-Arrival Estimation Algorithms for a Cubic Coprime Array

**DOI:** 10.3390/s22010136

**Published:** 2021-12-25

**Authors:** Pan Gong, Xixin Chen

**Affiliations:** College of Electronic Engineering, Nanjing Vocational University of Industry Technology, Nanjing 211106, China; chenxixin2002@sina.com

**Keywords:** direction-of-arrival estimation, massive multi-input multi-output, Cramer–Rao bound, ambiguity elimination, total array-based multiple signal classification, successive algorithm

## Abstract

In this paper, we investigate the problem of direction-of-arrival (DOA) estimation for massive multi-input multi-output (MIMO) radar, and propose a total array-based multiple signals classification (TA-MUSIC) algorithm for two-dimensional direction-of-arrival (DOA) estimation with a coprime cubic array (CCA). Unlike the conventional multiple signal classification (MUSIC) algorithm, the TA-MUSIC algorithm employs not only the auto-covariance matrix but also the mutual covariance matrix by stacking the received signals of two sub cubic arrays so that full degrees of freedom (DOFs) can be utilized. We verified that the phase ambiguity problem can be eliminated by employing the coprime property. Moreover, to achieve lower complexity, we explored the estimation of signal parameters via the rotational invariance technique (ESPRIT)-based multiple signal classification (E-MUSIC) algorithm, which uses a successive scheme to be computationally efficient. The Cramer–Rao bound (CRB) was taken as a theoretical benchmark for the lower boundary of the unbiased estimate. Finally, numerical simulations were conducted in order to demonstrate the effectiveness and superiority of the proposed algorithms.

## 1. Introduction

The multiple-input multiple-output (MIMO) radar [1,2,3] has received much attention in array signal processing, due to its capability to enhance spatial resolution, improve parameter identifiability, and achieve more degrees of freedom (DOFs) than traditional phased array radars [4]. Recently, a new MIMO system, called massive MIMO, has attracted great interest both in industry [5] and academia [6,7,8]. Compared to the traditional MIMO geometry, massive MIMO systems are equipped with a large number of antennae at the base station (BS), which brings a number of advantages: for example, higher throughput, improved spectral output, and enhanced link reliability [9,10,11,12,13].

Base stations rely on massive MIMO systems to uplink sounding signals, and compute channel knowledge to enable MIMO beam-forming [14,15,16]. Therefore, direction-of-arrival (DOA) estimation is important for massive MIMO], as it is needed to complete the predicted capacity gains [17,18]. Various DOA estimation algorithms have been proposed in recent years, such as improved multiple signal classification (MUSIC) [19], estimation of signal parameters via rotational invariance techniques (ESPRIT) [20], and the propagator method (PM) [21]. However, all these classic algorithms are conducted with a uniform array [22,23,24,25]. In a uniform array, inter-element spacing is limited to half a wavelength to avoid the problem of phase ambiguity [26].

In recent years, sparse arrays, such as nested [27] and coprime [28] arrays, have received considerable attention due to their capacity to improve estimation accuracy, enhance DOF, and mitigate the mutual coupling (MC) effect [29,30,31]. Practically speaking, a nested array is composed of at least one uniform linear array. For a uniform array, because the inter-element spacing is no larger than half a wavelength to avoid phase ambiguity, severe MC effects still occur. To reduce the MC effects, a general coprime linear array (CLA) which contains two uniform linear subarrays was proposed by [28]. The number of sensors of these two subarrays are defined as M and N. The inter-element spacing of the sensors is set to Nλ/2 and Mλ/2, respectively, where λ represents the wavelength of the carrier wave. The authors of [28] verified that a CLA with M+N−1 sensors can achieve O(MN) DOFs and improve the performance of the DOA estimation, where O(⋅) means the DOFs. A spatial smoothing technique-based algorithm was proposed in [31] which constructs a positive semi-definite covariance matrix to apply the subspace-based method. However, this algorithm involves a large number of snapshots to vectorize. In [32], a decomposed algorithm was proposed for CLA which separately processes the two subarrays by applying the conventional MUSIC algorithm and conducting a global angular spectral search. Meanwhile, the phase ambiguity problem is solved by combining the two estimates of two subarrays. A search-free algorithm was devised to estimate DOAs for coprime arrays with low computational complexity [33].

In practical applications, two-dimensional (2D) DOA estimation contains more important information. Consequently, a large number of investigations have been conducted into 2D DOA estimation generation [34,35,36]. The authors of [34] presented a general coprime planar array, proposing a partial spectral search algorithm to estimate 2D DOA which uses a linear relation to decrease the complexity. The authors of [35] proposed a combined ESPRIT algorithm which can achieve auto-paired angle estimation with lower complexity. The authors of [36] employed a reduced dimensional method which involved transforming the 2D spectral peak search to one-dimensional (1D) one within a small search sector. The authors of [37] proposed an initialization-based parallel factor method which utilized the PM to obtain initial estimates, then used these estimates to reconstruct the signal matrix; in this way, the complexity was significantly decreased. However, the above-mentioned studies, [34,35,36,37], are based on spectral peak searching, which results in high computational complexity.

For a two-dimensional array design, such methods can obtain superior performance in terms of azimuth angle estimation, but the elevation angle estimation is less good due to the fact that the array elements are distributed over a plane. From the perspective of improving performance in terms of the 2D DOA estimation of elevation angle, we propose considering the three-dimensional array model. Furthermore, the traditional antenna array is limited by the half-wavelength limit on inter-element spacing that avoids phase ambiguity. As a result, the array aperture is limited, which leads to poor resolution. Especially in the massive MIMO environment and in the context of 5G communication, which requires more antennae, the traditional array structure—with its limited array aperture, poorer resolution, lower estimation accuracy, and more severe MC effect—is not viable for practical applications. As a result, it is of great significance to expand the one-dimensional linear array and two-dimensional planar array models and introduce a three-dimensional coprime cubic array structure. Such designs can be used in massive MIMO, 5G communication, and UAVs. A coprime cubic array (CCA) geometry—a configuration which has been proven to outperform conventional uniform cubic arrays in terms of DOA estimation [38,39]—was proposed in [40]. In this paper, we exploit the same type of CCA configuration.

For CCA geometry, existing approaches, such as subspace-based algorithms [34,35,36], process the subarrays separately. This approach only employs the auto-covariance matrix of the subarray, resulting in achievable DOFs being damaged and making it impossible to solve the ambiguity problem caused by inter-element spacing larger than half a wavelength. In this paper, we propose a total array-based MUSIC algorithm (TA-MUSIC) for 2D DOA estimation, where not only the auto-covariance matrix, but also the mutual covariance matrix, is employed. Additionally, based on [34], we verified that, by using the coprime property, ambiguous estimates can be eliminated. To alleviate the computational burden, we present an ESPRIT-based MUSIC (E-MUSIC) algorithm which employs the computationally efficient ESPRIT algorithm to obtain initial DOA estimates, then conducts 1D spectral peak searches. Finally, numerical simulation results are presented to demonstrate the effectiveness of the proposed algorithms with CCA geometry.

The main contributions of our research are summarized as follows:(a)We propose a TA-MUSIC algorithm with CCA geometry to enable massive MIMOs to fully employ DOFs and achieve superior DOA estimation by employing both the auto-covariance matrix and the mutual covariance matrix of the entire array. In addition, we verify that by using the coprime property, the proposed algorithm can suppress the ambiguity problem.(b)We propose an E-MUSIC algorithm for 2D DOA estimation, which can effectively decrease the complexity of the classic MUSIC algorithm. After utilizing the ESPRIT algorithm to initialize and obtain a rough estimation, we then conduct a fine search within a smaller sector to achieve lower complexity.(c)Our numerical simulation results confirm that the proposed algorithms outperform the classical ESPRIT algorithm and the PM algorithm in DOA estimation.

The remaining sections of this paper are organized as follows. The array geometry and signal model are described in Section 2. The proposed algorithms are presented in Section 3. The complexity analysis is presented in Section 4, and the advantages of the proposed system discussed. Numerical simulations are provided in Section 5, and Section 6 discusses the conclusionsr.

Notations: We use lower-case bold characters for vectors and upper-case bold characters for matrices. The characters ${\left(\cdot \right)}^{T}$ and ${\left(\cdot \right)}^{H}$ denote the transpose and the conjugate transpose, respectively. The characters ⊙ and ⊗ represent the Khatri–Rao product and Kronecker product, respectively. The element diag(⋅) is a diagonal matrix which uses the elements of the matrix as its diagonal elements. The element E(⋅) is statistical expectation; $D_{m}(\cdot )$ is a diagonal matrix in which the *m*-th row of the matrix is employed; and angle(⋅) and arctan(⋅) are the phase operator and the arctangent function, respectively.

## 2. Signal Model

In this paper, we employ a CCA configuration which can further enlarge the array aperture and improve DOA estimation performance [40].

As described in [40], a CCA configuration incorporates two uniform cubic subarrays. One subarray is set with $M_{1}\times T_{1}\times P_{1}$ sensors. The other is set with $M_{2}\times T_{2}\times P_{2}$ sensors. For subarray 1, the inter-element spacing is determined as $d_{x1}=M_{2}\lambda /2$ for the x-axis direction, $d_{y1}=$$T_{2}\lambda /2$ for the y-axis direction, and $d_{z1}=P_{2}\lambda /2$ for the z-axis direction, where λ represents the wavelength. In subarray 2, the inter-element spacing is $d_{x2}=M_{1}\lambda /2$, $d_{y2}=T_{1}\lambda /2$ and $d_{z2}=P_{1}\lambda /2$, for the x-, y-, and z-axes, respectively. The total number of the sensors is therefore computed according to the equation $T_{CCA}=M_{1}T_{1}P_{1}+M_{2}T_{2}P_{2}-1$. Figure 1 depicts an example of a CCA configuration where $M_{1}=T_{1}=P_{1}=4$ and $M_{2}=T_{2}=P_{2}=3$.

Suppose that *K* far-field uncorrelated narrowband incident signals impinge on CCA with angles $\left\{\left(\theta_{k},\phi_{k}\right)|k=1,2,\ldots ,K\right\}$, where $\theta_{k}$ represents the elevation angle and $\phi_{k}$ denotes the azimuth angle of the *k*-th signal; $K<\min\{M_{1}T_{1}P_{1},M_{2}T_{2}P_{2}\}$; $\theta_{k}\in (0{}^{\circ},90{}^{\circ})$; and $\phi_{k}\in$(0°,180°). For simplicity, we set $u_{k}=\sin\theta_{k}\cos\phi_{k}\in (-1,1),$$v_{k}=\sin\theta_{k}sin\phi_{k}\in (0,1)$ and $w_{k}=$$\cos\theta_{k}\in (0,1)$. Then, we take subarray *i* as an example to illustrate the proposed algorithm in the following part. Accordingly, the subarray has $M_{i}\times T_{i}\times P_{i}(i=1,2)$.

If we denote the outputs of two subarrays as per [34], then:(1)X1=A1S+N1
(2)X2=A2S+N2
where S=[s1,s2,…,sK]T∈ℂK×L represents the signal matrix and L denotes the number of snapshots. Ni∈ℂMiTiPi×L(i=1,2) is the additive Gaussian white noise matrix. The variance and mean are defined as σn2 and zero, respectively. Ai∈ℂMiTiPi×K is the directional matrix of the *i*-th subarray with Mi×Ti×Pi(i=1,2) sensors and is presented by Ai=[ai(u1,v1,w1),ai(u2,v2,w2),…,ai(uK,vK,wK)] and ai(uk,vk,wk)=ax_i(uk)⊗ay_i(vk)⊗az_i(wk),(k=1,2,…,K) denotes the steering vector.
(3)ax_i(uk)=[1,ej2πdxiuk/λ,…,ej2π(Mi−1)dxiuk/λ]T,
(4)ay_i(vk)=[1,ej2πdyivk/λ,…,ej2π(Ti−1)dyivk/λ]T,
(5)az_i(wk)=[1,ej2πdziwk/λ,…,ej2π(Pi−1)dziwk/λ]T,

The directional matrix can also be denoted as:(6)$$\begin{array}{ll} A_{i} & =A_{x\_i}⊙A_{y\_i}⊙A_{z\_i}\\ & =\left[\begin{array}{l} \;A_{z\_i}D_{1}(A_{y\_i})D_{1}(A_{x\_i})\\ \;A_{z\_i}D_{1}(A_{y\_i})D_{2}(A_{x\_i})\\ \;\;\;\;\vdots \\ A_{z\_i}D_{T_{i}}(A_{y\_i})D_{M_{i}}(A_{x\_i}) \end{array}\right] \end{array}$$

Here, we use $A_{x\_i},A_{y\_i},A_{z\_i}$ as directional matrices of subarray *i*. $A_{x\_i}$ represents the x-axis, $A_{y\_i}$ denotes the y-axis, and $A_{z\_i}$ is the z-axis. The character ⊙ denotes the Khatri–Rao product, and $D_{m}(\cdot )$ denotes a diagonal matrix which employs the *m*-th row values of the matrix.

## 3. Proposed Method for DOA Estimation

In this section, we examine the classic MUSIC algorithm [32] for CCA geometry. We then propose a TA-MUSIC algorithm to solve the problems encountered with the classic MUSIC algorithm. Note that the TA-MUSIC algorithm involves spectral peak searching within the global angular sector, generating a high level of computational complexity; we therefore propose using an E-MUSIC algorithm to alleviate the computational burden. To be more explicit, we take subarray 1 as an example to explain the proposed algorithm.

### 3.1. Review

In [32], the authors ran the subarrays of CCA separately. Correspondingly, the covariance matrix for output signal of the subarray *i* can be acquired using the formula:(7)$${\hat{R}}_{i}=(1/L)\sum_{l=1}^{L}X_{i}X_{i}^{H}$$

Therefore, according to eigenvalue decomposition (EVD), we have:(8)$${\hat{R}}_{i}={\hat{E}}_{s,i}{\hat{D}}_{s,i}{\hat{E}}_{s,i}^{H}+{\hat{E}}_{n,i}{\hat{D}}_{n,i}{\hat{E}}_{n,i}^{H}$$
where ${\hat{D}}_{s,i}$ represents the largest *K* eigenvalues in the signal subspace and ${\hat{E}}_{s,i}$ represents the corresponding signal subspace matrix. The noise subspace, which is composed of remaining eigenvalues, is denoted by ${\hat{D}}_{n,i}$ and the corresponding noise subspace matrix is ${\hat{E}}_{n,i}$.

According to the 1D MUSIC spectrum [19], we can denote the 3D MUSIC spatial spectral function for subarray *i* as
(9)$$f_{i}(u,v,w)=\frac{1}{a_{i}^{H}(u,v,w)E_{n,i}E_{n,i}^{H}a_{i}(u,v,w)}(i=1,2)$$
where $a_{i}(u,v,w)=a_{x,i}^{}(u)⊗a_{y,i}^{}(v)⊗a_{z,i}^{}(w)$, u=sinθcosϕ,v=sinθsinϕ and w=cosθ.

### 3.2. TA-MUSIC Algorithm

Unlike the algorithms that conduct the subarrays separately [32,33,34], in the proposed TA-MUSIC algorithm, both covariance matrices of the total array are employed to estimate DOAs by stacking the outputs of the two subarrays, as follows:(10)$$\begin{array}{ll} X & =\left[\begin{array}{l} X_{1}\\ X_{2} \end{array}\right]=\left[\begin{array}{l} A_{1}\\ A_{2} \end{array}\right]S+\left[\begin{array}{l} N_{1}\\ N_{2} \end{array}\right]\\ & =AS+N \end{array}$$
where $A={\left[A_{1}^{T},A_{2}^{T}\right]}^{T}$ and $n(t)={\left[n_{1}^{T}(t),n_{2}^{T}(t)\right]}^{T}$. From this, the covariance matrix of the total array can be estimated using the formula:(11)$$\hat{R}=(1/L)\sum_{l=1}^{L}XX^{H}$$

By operating EVD, we have:(12)$$\hat{R}={\hat{E}}_{s}{\hat{D}}_{s}{\hat{E}}_{s}^{H}+{\hat{E}}_{n}{\hat{D}}_{n}{\hat{E}}_{n}^{H}$$
where ${\hat{E}}_{s}$ is the signal subspace matrix and ${\hat{E}}_{n}$ denotes the noise subspace matrix. ${\hat{D}}_{s}$ and ${\hat{D}}_{n}$ denote matrixes which contain the largest *K* eigenvalues and the remaining eigenvalues, respectively.

Then, according to the orthogonality that exists between the signal subspace and noise subspace, we can denote the spatial spectral function as:(13)$$f(u,v,w)=\frac{1}{a^{H}(u,v,w)E_{n}E_{n}^{H}a(u,v,w)}$$
where $a(u,v,w)={\left[a_{1}^{T}(u,v,w)\;\;a_{2}^{T}(u,v,w)\right]}^{T}$. After searching over $\theta_{k}\in (0{}^{\circ},90{}^{\circ})$ and $\phi_{k}\in (0{}^{\circ},180{}^{\circ})$, the TA-MUSIC spectrum can be obtained without any ambiguous phase problems arising. This is demonstrated in Lemma 1, below [34]:

**Lemma** **1.***We suppose*$\left(\theta_{k}^{},\phi_{k}^{}\right)$*to be the real DOA for the k-th signal. In addition, only one DOA estimate pair*$\left({\hat{\theta }}_{k},{\hat{\phi }}_{k}\right)$*generates a spectral peak in the TA-MUSIC spectrum. Therefore,*$\left({\hat{\theta }}_{k},{\hat{\phi }}_{k}\right)$*is the estimates of the real DOA*$\left(\theta_{k},\phi_{k}\right)$.

**Proof.** There is another DOA estimate, $\left({\hat{\theta }}_{k}^{a},{\hat{\phi }}_{k}^{a}\right)$, besides $\left({\hat{\theta }}_{k}^{},{\hat{\phi }}_{k}^{}\right)$ and $\left({\hat{\theta }}_{k}^{a},{\hat{\phi }}_{k}^{a}\right)$, which can be used to generate a spectral peak for the real DOA $\left(\theta_{k},\phi_{k}\right)$. In other words, this means:
(14)$$a^{H}({\hat{u}}_{k},{\hat{v}}_{k},{\hat{w}}_{k})E_{n}=a^{H}({\hat{u}}_{k}^{a}{}_{},{\hat{v}}_{k}^{a},{\hat{w}}_{k}^{a})E_{n}$$
(15)$$a({\hat{u}}_{k},{\hat{v}}_{k},{\hat{w}}_{k})=a({\hat{u}}_{k}^{a}{}_{},{\hat{v}}_{k}^{a},{\hat{w}}_{k}^{a})$$
where $a({\hat{u}}_{k},{\hat{v}}_{k},{\hat{w}}_{k})={\left[a_{1}^{T}({\hat{u}}_{k},{\hat{v}}_{k},{\hat{w}}_{k}),a_{2}^{T}({\hat{u}}_{k},{\hat{v}}_{k},{\hat{w}}_{k})\right]}^{T},\;a({\hat{u}}_{k}^{a},{\hat{v}}_{k}^{a},{\hat{w}}_{k}^{a})={\left[a_{1}^{T}({\hat{u}}_{k}^{a},{\hat{v}}_{k}^{a},{\hat{w}}_{k}^{a}),a_{2}^{T}({\hat{u}}_{k}^{a},{\hat{v}}_{k}^{a},{\hat{w}}_{k}^{a})\right]}^{T},$ $a_{i}({\hat{u}}_{k},{\hat{v}}_{k},{\hat{w}}_{k})=a_{x\_i}({\hat{u}}_{k},{\hat{v}}_{k},{\hat{w}}_{k})⊗a_{y\_i}({\hat{u}}_{k},{\hat{v}}_{k},{\hat{w}}_{k})⊗a_{z\_i}({\hat{u}}_{k},{\hat{v}}_{k},{\hat{w}}_{k}),a_{i}({\hat{u}}_{k}^{a},{\hat{v}}_{k}^{a},{\hat{w}}_{k}^{a})=a_{x\_i}^{}({\hat{u}}_{k}^{a},{\hat{v}}_{k}^{a},{\hat{w}}_{k}^{a})⊗$
$a_{y\_i}^{}({\hat{u}}_{k}^{a},{\hat{v}}_{k}^{a},{\hat{w}}_{k}^{a})⊗a_{z,i}^{}({\hat{u}}_{k}^{a},{\hat{v}}_{k}^{a},{\hat{w}}_{k}^{a})(i\in [1,2]),$ ${\hat{u}}_{k}=\sin{\hat{\theta }}_{k}\cos{\hat{\phi }}_{k},$ ${\hat{u}}_{k}^{a}=\sin\theta_{k}^{a}\cos\phi_{k}^{a},$ ${\hat{v}}_{k}=\sin{\hat{\theta }}_{k}\sin{\hat{\phi }}_{k},{\hat{v}}_{k}^{a}=$ $\sin{\hat{\theta }}_{k}^{a}$ $\sin{\hat{\phi }}_{k}^{a}$, ${\hat{w}}_{k}=\cos\theta_{k}$ and ${\hat{w}}_{k}^{a}=\cos{\hat{\theta }}_{k}^{a}$.Then we have:
(16)$$a_{x\_i}({\hat{u}}_{k},{\hat{v}}_{k},{\hat{w}}_{k})=a_{x\_i}^{}({\hat{u}}_{k}^{a},{\hat{v}}_{k}^{a},{\hat{w}}_{k}^{a})$$
(17)$$a_{y\_i}({\hat{u}}_{k},{\hat{v}}_{k},{\hat{w}}_{k})=a_{y\_i}^{}({\hat{u}}_{k}^{a},{\hat{v}}_{k}^{a},{\hat{w}}_{k}^{a})$$
(18)$$a_{z\_i}({\hat{u}}_{k},{\hat{v}}_{k},{\hat{w}}_{k})=a_{z\_i}^{}({\hat{u}}_{k}^{a},{\hat{v}}_{k}^{a},{\hat{w}}_{k}^{a})$$Specifically,
(19)$${\hat{u}}_{k}-{\hat{u}}_{k}^{a}=\frac{2k_{ui}\pi }{M_{j}},{\hat{v}}_{k}-{\hat{v}}_{k}^{a}=\frac{2k_{vi}\pi }{T_{j}},{\hat{w}}_{k}-{\hat{w}}_{k}^{a}=\frac{2k_{wi}\pi }{P_{j}}$$ where $k_{ui}\in (-M_{j},M_{j})$, $k_{vi}\in (-T_{j}/2,T_{j}/2)$, $k_{wi}\in$
$\left(-P_{j}/2,P_{j}/2\right)$ and i,j∈{1,2},
i≠j.Then we have
(20)$$\frac{k_{u1}}{M_{2}}=\frac{k_{u2}}{M_{1}},\frac{k_{v1}}{T_{2}}=\frac{k_{v2}}{T_{1}},\frac{k_{w1}}{P_{2}}=\frac{k_{w2}}{P_{1}}$$Since *M_1_*, *M_2_*,
$T_{1}$,$T_{2}$ and $P_{1},P_{2}$ are coprime integers pairs, respectively, Equation (20) can hold only in the case of $k_{u1}=k_{u2}=0,k_{v1}=k_{v2}=0,k_{w1}=k_{w2}=0$. This indicates that no ambiguity problem has arisen. □

As described above, the proposed algorithm enables spectral searching within the global angular sector, which results in expensive computational complexity. In the following section, we propose using an E-MUSIC algorithm to significantly reduce the computational complexity.

### 3.3. The E-MUSIC Algorithm

From Equation (1), we know that the subarray with $M_{i}\times T_{i}\times P_{i}$ sensors has uniform inter-element spacing and the properties of a Vandermonde matrix, indicating that the ESPRIT algorithm, which has high computational efficiency, can be employed to achieve initial DOA estimates.

For subarray 1, with $M_{1}\times T_{1}\times P_{1}$ sensors, according to Equation (7) the covariance matrix can be computed as follows:(21)$${\hat{R}}_{1}=(1/L)\sum_{l=1}^{L}X_{1}X_{1}^{H}$$

Then, according to Equation (8), we can perform the eigenvalue decomposition of ${\hat{R}}_{1}$ as follows:(22)$${\hat{R}}_{1}={\hat{E}}_{s,1}{\hat{D}}_{s,1}{\hat{E}}_{s,1}^{H}+{\hat{E}}_{n,1}{\hat{D}}_{n,1}{\hat{E}}_{n,1}^{H}$$

In the noiseless scenery, ${\hat{E}}_{s,1}$ satisfies
(23)$${\hat{E}}_{s,1}=A_{1}T$$
where **T** is a nonsingular matrix and ${\hat{E}}_{s,1}$ is signal subspace which is segmented into ${\hat{E}}_{x1,1}$ and ${\hat{E}}_{x2,1}$, denoting ${\hat{E}}_{s,1}(1:(M_{1}-1)T_{1}P_{1},:)$ and ${\hat{E}}_{s,1}(T_{1}P_{1}+1:T_{1}M_{1}P_{1},:)$; and corresponding steering matrices $A_{x1,1}$ and $A_{x2,1}$, denoting $A_{1}(1:(M_{1}-1)T_{1}P_{1},:)$ and $A_{1}(T_{1}P_{1}+1:M_{1}T_{1}P_{1},:)$, respectively. Based on the uniformities of steering matrices, we have:(24)$$A_{x2,1}=A_{x1,1}\Phi_{x,1}$$
where $\Phi_{x,1}=diag(e^{-j2\pi d{\hat{u}}_{1}^{a}/\lambda },e^{j2\pi d{\hat{u}}_{2}^{a}/\lambda },\ldots ,$$e^{j2\pi d{\hat{u}}_{K}^{a}/\lambda })$. The signal subspace then satisfies [27]:(25)$$\left[\begin{array}{l} {\hat{E}}_{x1,1}\\ {\hat{E}}_{x2,1} \end{array}\right]=\left[\begin{array}{l} A_{x1,1}T\\ A_{x2,1}T \end{array}\right]=\left[\begin{array}{l} \;A_{x1,1}T\\ A_{x1,1}\Phi_{x,1}T \end{array}\right]$$

Then we have:(26)$${\hat{E}}_{x1,1}^{÷}{\hat{E}}_{x2,1}=T^{-1}\Phi_{x,1}T$$

By performing eigenvalue decomposition (EVD) on covariance matrix $\Psi_{x1}={\hat{E}}_{x1,1}^{÷}{\hat{E}}_{x2,1}$ we can obtain *K* estimates ${\hat{u}}_{k}^{a},(k=1,2,\ldots ,K)$, and using the eigenvectors, we can obtain an estimated T, which is employed to achieve automatically paired angles in the following part.

Next, we rearrange the rows of ${\hat{E}}_{s,1}$ to obtain a new signal subspace, ${\hat{E}}_{s,1}^{\prime }$
(27)$${\hat{E}}_{s,1}^{\prime }=A_{1}^{\prime }T=\left(A_{y,1}⊙A_{x,1}⊙A_{z,1}\right)T=\left(A_{y,1}⊙A_{xz,1}\right)T$$

Similarly to ${\hat{E}}_{s,1}$, we then segment ${\hat{E}}_{s,1}^{\prime }$ into ${\hat{E}}_{y1,1}^{}$ and ${\hat{E}}_{y2,1}^{}$, denoting ${\hat{E}}_{s,1}^{\prime }(1:(T_{1}-1)M_{1}P_{1},:)$ and ${\hat{E}}_{s,1}^{\prime }(M_{1}P_{1}:T_{1}M_{1}P_{1},:)$, respectively. Then we perform EVD on $\Psi_{y1}={\hat{E}}_{y1,1}^{÷}{\hat{E}}_{y1,2}$ to obtain $\Phi_{y,1}$ and ${\hat{v}}_{k}^{a}$. Likewise, we can achieve $\Phi_{z,1}$ and ${\hat{w}}_{k}^{a}$ by applying the same algorithm. In this part of the calculation, in order to decrease the complexity, we chose to explore only two variants. In the process of computing $\Phi_{x,1}$,$\Phi_{y,1}$, and $\Phi_{z,1}$, estimated DOAs $\left({\hat{u}}_{k}^{a},{\hat{v}}_{k}^{a}\right)$ with automatic pairing can be obtained using the same matrix, T [28]. According to Equation (19), if the inter-element spacing is larger than half a wavelength, the phase ambiguity problem arises and all the DOA estimates we obtain will be ambiguous. However, according to the coprime property, the phase ambiguity problem can be eliminated and unambiguous values can be attained, as demonstrated in Lemma 1. To obtain more accurate DOA estimates, the unambiguous values $\left({\hat{u}}_{k}^{ini},{\hat{v}}_{k}^{ini}\right)$ are employed to initialize the spatial spectral function in Equation (13).

Unlike the TA-MUSIC algorithm, which generates high computational complexity due to its use of 2D spectral peak searching within a global angular sector of $\theta_{k}\in (0{}^{\circ},90{}^{\circ}),\phi_{k}\in (0{}^{\circ},180{}^{\circ})$, the proposed E-MUSIC algorithm transforms the 2D spectral peak search into a 1D spectral peak search over the sector, $u_{k}\in (-1,1),v_{k}\in (-1,1)$, which is significantly more computationally efficient.

For the *k*-th signal, the initial estimate $\left({\hat{u}}_{k}^{ini},{\hat{v}}_{k}^{ini}\right)$ is initially employed. From this, a new 1D spatial spectral function can be obtained:(28)$$f_{w}({\hat{u}}_{k}^{ini},{\hat{v}}_{k}^{ini},w)=\frac{1}{a^{H}({\hat{u}}_{k}^{ini},{\hat{v}}_{k}^{ini},w)E_{n}E_{n}^{H}a({\hat{u}}_{k}^{ini},{\hat{v}}_{k}^{ini},w)}$$

A search is conducted over a small sector w∈(0,1), and a 1D spatial spectral peak search is then performed to achieve an improved DOA estimate for $w_{k}$. Similarly, by using $w_{k}$ and ${\hat{v}}_{k}^{ini}$, we can construct the 1D spatial spectral function of u as:(29)$$f_{u}(u,{\hat{v}}_{k}^{ini},{\hat{w}}_{k})=\frac{1}{a^{H}(u,{\hat{v}}_{k}^{ini},{\hat{w}}_{k})E_{n}E_{n}^{H}a(u,{\hat{v}}_{k}^{ini},{\hat{w}}_{k})}$$

From this, it is possible to obtain more accurate estimates from a 1D spectral peak search. The final estimates can be obtained by means of:(30)$$\left\{ \begin{array}{l} {\hat{\theta }}_{k}=\arccos{\hat{w}}_{k}\\ {\hat{\phi }}_{k}=\arccos({\hat{u}}_{k}/\sin(\arccos{\hat{w}}_{k})) \end{array}\right.$$

### 3.4. Detailed Steps

The detailed steps of the E-MUSIC algorithm are provided as follows.

Step 1. Compute the covariance matrix $\hat{R}$ of the total array according to Equation (8), then decompose $\hat{R}$ to attain the covariance matrix $\hat{R}$ of subarray 1.

Step 2. Conduct an EVD of $\hat{R}$ and ${\hat{R}}_{1}$. The corresponding noise and signal subspaces are denoted as ${\hat{E}}_{n}$ and ${\hat{E}}_{s,1}$ by Equations (19) and (22), respectively.

Step 3. Apply the ESPRIT algorithm to ${\hat{E}}_{s,1}$. Then, use Equations (20), (26) and (27) to obtain the initial estimates $({\hat{u}}_{k}^{ini},{\hat{v}}_{k}^{ini}),k=1,2,\ldots ,K$.

Step 4. For the *k*-th signal, to achieve a more accurate estimate ${\hat{w}}_{k}$, generate the 1D spatial spectral function for w∈(0,1) with $\left({\hat{u}}_{k}^{ini},{\hat{v}}_{k}^{ini}\right)$, where k=1,2,…,K.

Step 5. To achieve a more accurate estimate, ${\hat{u}}_{k}$, generate the 1D spatial spectral function for u∈(−1,1) with $\left(w_{k}^{},{\hat{v}}_{k}^{ini}\right)$k=1,2,…,K.

Step 6. Use Equation (30) to obtain the final DOA estimate $\left({\hat{\theta }}_{k},{\hat{\phi }}_{k}\right)$k=1,2,…,K.

## 4. Performance Analysis

### 4.1. Computational Complexity

In this subsection, we compare the complexity of the proposed TA-MUSIC algorithm, the E-MUSIC algorithm, and the classic MUSIC algorithm [32] using the same CCA for each. For the TA-MUSIC algorithm, the total complexity is $G^{2}L+G^{3}+(180/\tau )(90/\tau )$[G(G−K)], where *L* represents the number of snapshots, $G=G_{1}+G_{2},G_{1}=$$M_{1}T_{1}P_{1},G_{2}=$$M_{2}T_{2}P_{2}$, and τ denotes the searching step. For the E-MUSIC algorithm, the total complexity is $G_{}^{2}L+G^{3}+$(τ/3)G(G−K).

For clarity, we list the computational complexity of the mentioned algorithms in Table 1, where $M_{1}=4$, $T_{1}=4$, $P_{1}=4$, $M_{2}=3$, $T_{2}=3$, $P_{2}=3$, L=200, K=2 and the searching step τ is set to 0.02. We can clearly see that the proposed TA-MUSIC algorithm has the almost the same complexity as the TSS algorithm, while the proposed E-MUSIC algorithm has the lowest computational complexity compared to the others.

Figure 2 compares the complexity with increasing number of sensors of the three algorithms, classic MUSIC, TA-MUSIC and E-MUSIC. From this chart, we can observe that the computational complexity of these three algorithms increases with the number of sensors. It can also be seen that E-MUSIC has the lowest complexity among the three. Figure 3 compares the complexity of the three algorithms as the number of snapshots increases. This chart indicates E-MUSIC’s superiority due to lower complexity. Figure 4 compares the complexity under the variant searching step. It clearly indicates that the computational complexity increases as searching decreases.

### 4.2. Degree of DOF

The proposed TA-MUSIC algorithm can obtain $M_{1}T_{1}P_{1}+M_{2}T_{2}P_{2}-1$ DOFs, which indicates TA-MUSIC can detect $M_{1}T_{1}P_{1}+M_{2}T_{2}P_{2}-2$ target signals by using $M_{1}T_{1}P_{1}+$$M_{2}T_{2}P_{2}-1$ sensors.
(31)$${\mathrm{DOF}}_{\mathrm{CCA}}^{\mathrm{TA}-\mathrm{MUSIC}}=M_{1}T_{1}P_{1}+M_{2}T_{2}P_{2}-1$$

The algorithms in [34,35,36] divide the array into two subarrays, then estimate DOA separately. In this way, much of the array information is not fully employed, resulting in a loss of DOFs. The method described in [34,35,36] can achieve a greater number of DOFs, as
(32)$${\mathrm{DOF}}_{\mathrm{CCA}}^{\mathrm{methods}\mathrm{in}}=\min\left(M_{1}T_{1}P_{1},M_{2}T_{2}P_{2}\right)$$
where $M_{1}=T_{1}=P_{1}>M_{2}=$ $T_{2}=P_{2}$.

Specifically, the E-MUSIC algorithm has $M_{2}T_{2}P_{2}$ DOFs,
(33)$${\mathrm{DOF}}_{\mathrm{CCA}}^{E-\mathrm{MUSIC}}=\max\left(M_{1}T_{1}P_{1},M_{2}T_{2}P_{2}\right)$$

Thus, the proposed algorithms can detect many more signals than conventional algorithms with the same sensors. From Equations (31)–(33), we can describe the relationship of DOFs of the three algorithms as:(34)$${\mathrm{DOF}}_{\mathrm{CCA}}^{\mathrm{methods}\mathrm{in}}{<\mathrm{DOF}}_{\mathrm{CCA}}^{E-\mathrm{MUSIC}}{<\mathrm{DOF}}_{\mathrm{CCA}}^{\mathrm{TA}-\mathrm{MUSIC}}$$

### 4.3. Advantages

The performance of the TA-MUSIC and E-MUSIC algorithms are summarized as follows:

The proposed TA-MUSIC performs better DOA estimations by employing all of the array information, including the auto-covariance matrix and the mutual covariance matrix, whereas E-MUSIC only utilizes the auto-covariance matrix information. In addition, TA-MUSIC can fully achieve DOFs of $M_{1}T_{1}P_{1}+M_{2}T_{2}P_{2}-1$, while the algorithms in [34,35,36] only achieve $M_{2}T_{2}P_{2}$ DOFs. Furthermore, E-MUSIC can obtain $M_{1}T_{1}P_{1}$ DOFs, which are larger than $M_{2}T_{2}P_{2}(M_{1}>M_{2})$.The proposed E-MUSIC algorithm significantly reduces computational complexity by transforming the 2D spatial spectral peak search into a 1D one, whereas the algorithms in [34,35,36] and TA-MUSIC require 2D spectral peak searching.The proposed TA–MUSIC algorithm can attain paired angles automatically and outperforms the conventional ESPRIT and PM algorithms in DOA estimation performance.

### 4.4. Cramer–Rao Bound

In this part, we provide the Cramer–Rao bound (CRB) [41] derivation of DOA estimation with a CCA. The signal covariance matrix is achieved through $R_{s}=SS^{H}/L$.

According to [41], the CRB with a CCA can be obtained using the following formula:(35)$$C_{\theta ,\phi }=\frac{\sigma_{}^{2}}{2L}{\left\{Re\left[D_{}^{H}\prod_{A}^{⊥}D⊕{\hat{R}}^{T}\right]\right\}}^{-1}$$
where $D=[\partial a_{1}/\partial \theta_{1},\partial a_{2}/\partial \theta_{2},\ldots ,\partial a_{K}/\partial \theta_{K},$
$\partial a_{1}/\partial \phi_{1},\partial a_{2}/\partial \phi_{2},\ldots ,\partial a_{K}/\partial \phi_{K}]$, $\prod_{A}^{⊥}=I_{N_{1}M_{1}J_{1}+N_{2}M_{2}J_{2}}\;-$ $A{\left(A_{}^{H}A\right)}^{-1}A_{}^{H}$ and $a_{k}$ are the *k*-th column of A(k=1,2,…,K), $\hat{R}=[{\left(R_{s},R_{s}\right)}^{T},$ ${\left(R_{s},R_{s}\right)}^{T}{]}^{T}$.

## 5. Simulation Results

In this section, we provide numerical simulation results to demonstrate the estimation performance of the proposed algorithms with a CCA. The CCA configuration consisted of two UCA configurations of 4×4×4 and 3×3×3 sensors respectively, where $d_{x1}=3\lambda /2$, $d_{y1}=3\lambda /2$, $d_{z1}=3\lambda /2$, $d_{x2}=4\lambda /2$, $d_{y2}=4\lambda /2$ and $d_{z2}=4\lambda /2$. Suppose that K=2 signals impinge on the CCA from $\left(\theta_{1},\phi_{1})=(10^{\circ },20^{\circ }\right)$ and $\left(\theta_{2},\phi_{2})=(20^{\circ },30^{\circ }\right)$. Figure 5 and Figure 6 are the scattering images for TA-MUSIC and E-MUSIC, respectively. As they show, both the proposed algorithms were able to successfully detect the target signals.

The root mean square error (RMSE) was exploited as the estimation performance metric, defined by the following formula:(36)$$RMSE=\sqrt{\frac{1}{PK}\sum_{p=1}^{P}\sum_{k=1}^{K}\left({\left(\theta_{k}-{\hat{\theta }}_{k}^{p}\right)}^{2}+{\left(\phi_{k}-{\hat{\phi }}_{k}^{p}\right)}^{2}\right)}$$
where P represents Monte-Carlo simulation iterations and K denotes the number of target signals. $\left({\hat{\theta }}^{p}{}_{k},{\hat{\phi }}_{k}^{p})\;(k=1,2\ldots ,K\right)$ represents the estimate of the *k*-th angle $\left(\theta_{k},\phi_{k}\right)$ for the *p*-th trial. For this study, P=500.

### 5.1. Comparison of the DOA Estimation Performance of Different Algorithms

In this subsection, we present the RMSE performance of various algorithms with a CCA, specifically, the PM [42] algorithm, the ESPRIT [43] algorithm, and the proposed TA-MUSIC and E-MUSIC algorithms. It is assumed that there are two DOAs from the directions of $\left(\theta_{1},\phi_{1})=(10^{\circ },20^{\circ }\right)$ and $(\theta_{2},\phi_{2})=$ $\left(20^{\circ },30^{\circ }\right)$. Figure 7 depicts the RMSE against the SNR (signal-to-noise ratio), where the number of snapshots is *L* = 200. Figure 8 depicts the RMSE against the number of snapshots, where SNR = 0dB. In Figure 7, it can be seen that estimation performance improved as with increasing SNR, and that the proposed TA-MUSIC algorithm outperforms the others, because it utilizes all the available information, from both the auto-covariance matrix and the mutual-covariance matrix. In addition, Figure 7 and Figure 8 show that the E-MUSIC algorithm, which uses a successive scheme to estimate the signals’ direction of arrival, can achieve prominent DOA estimation performance, along with the high computational efficiency demonstrated in Figure 2 and Figure 3.

### 5.2. RMSE with a Varying Number of Sensors

In this subsection, we illustrate how DOA estimation performance changes with the number of snapshots for the proposed E-MUSIC algorithm with CCA geometry which consists of two subarrays. The first subarray, $M_{1}\times N_{1}\times J_{1}$, was simulated with various geometries, where the number of sensors was $M_{1}=N_{1}=$$J_{1}=[4,5,7]$. The other subarray remained set to $M_{2}\times N_{2}\times J_{2}=3\times 3\times 3$ sensors throughout. Figure 9 depicts RMSE performance with increasing SNR levels for the various-sized sensor arrays, where *L* = 200. Figure 10 shows RMSE performance as the number of snapshots increases, where SNR = 0dB. As can be seen in Figure 9, DOA estimation performance improved as the SNR increased, due to the decrease in noise. Figure 10 clearly shows that the DOA estimation performance of the E-MUSIC algorithm improved with the number of snapshots.

### 5.3. Resolution Performance

If we assume that two target signals are closely located and impinge on the CCA array, and define the two signals as $\left({\hat{\theta }}_{1},{\hat{\phi }}_{1}\right)$ and $\left({\hat{\theta }}_{2},{\hat{\phi }}_{2}\right)$, then these two signals are detected if $\left|\theta_{}-\hat{\theta }\right|<\left|\theta_{1}-\theta_{2}\right|/2$ and $\left|\phi -\hat{\phi }\right|<\left|\phi_{1}-\phi_{2}\right|/2$. We selected two closely located signals of $\left(\theta_{1},\theta_{2})=(10^{\circ },11^{\circ }\right)$ and $\left(\phi_{1},\phi_{2})=(30^{\circ },32^{\circ }\right)$ to study the estimation probability of the proposed algorithms. Figure 11 and Figure 12 show the estimation probability of the proposed TA-MUSIC and E-MUSIC algorithms and the conventional ESPRIT and PM algorithms according to SNR and number of snapshots, respectively. As depicted in these two figures, the resolution performance improved with increasing SNR and number of snapshots. Specifically, from Figure 11, it can be seen that the DOA estimation improved with the improved SNR, and Figure 12 clearly shows that the DOA estimation performance improved with the number of snapshots. It can also be seen that the conventional ESPRIT and PM algorithms presented almost identical DOA estimation performances, but the proposed TA-MUSIC and E-MUSIC algorithms demonstrated far better estimation performances.

## 6. Conclusions

In this paper, we propose using a TA-MUSIC algorithm for 2D DOA estimation with a CCA configuration. Unlike the conventional MUSIC algorithms which only utilize information from the auto-covariance matrix, the TA-MUSIC algorithm employs all the available information from both this and the mutual-covariance matrix. This enables the full achievement of the DOF and improves DOA estimation performance. Moreover, in order to reduce computational complexity, we propose using the computationally efficient E-MUSIC for 2D DOA estimation. This algorithm exploits the ESPRIT algorithm to obtain an initial estimate. The simulation results verified that the proposed algorithms demonstrate superior DOA estimation performance compared to the conventional algorithms.

## Figures and Tables

**Figure 1 sensors-22-00136-f001:**
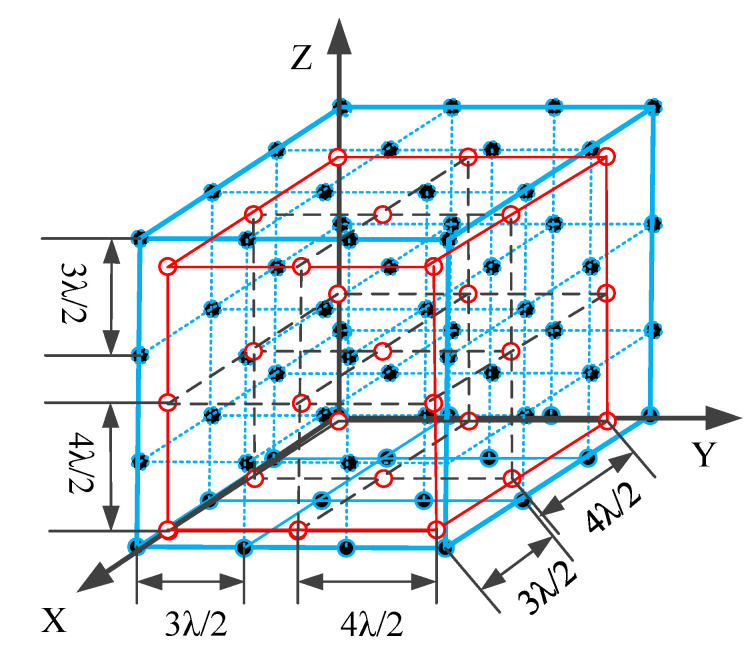
The structure of a CCA configuration ($M_{1}=4,T_{1}=4,P_{1}=4$ and $M_{2}=3,T_{2}=3,P_{2}=3$ ).

**Figure 2 sensors-22-00136-f002:**
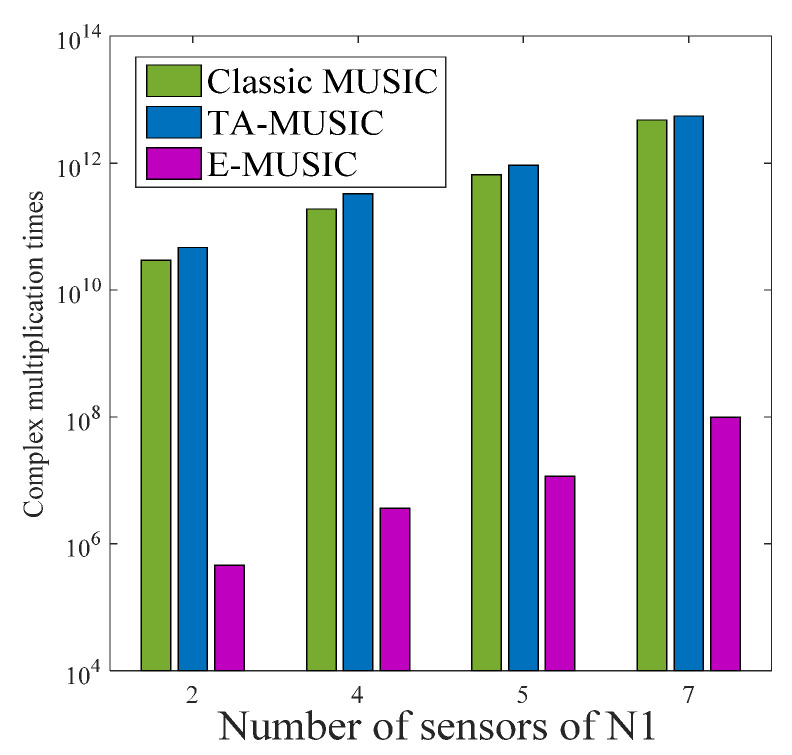
Algorithm complexity according to number of sensors.

**Figure 3 sensors-22-00136-f003:**
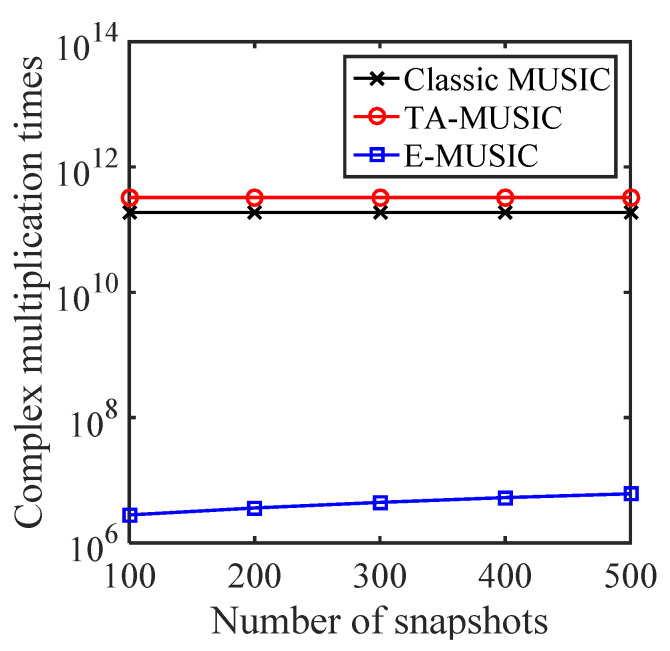
Algorithm complexityaccording to number of snapshots.

**Figure 4 sensors-22-00136-f004:**
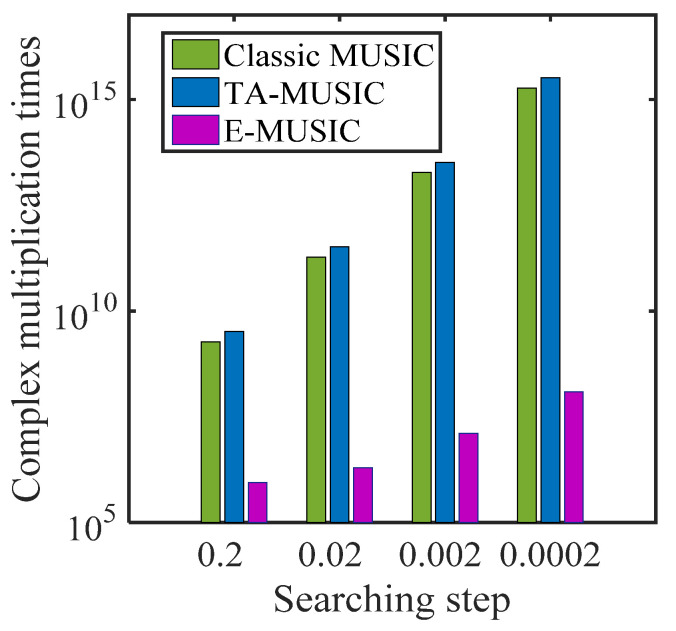
Algorithm complexity according to searching step.

**Figure 5 sensors-22-00136-f005:**
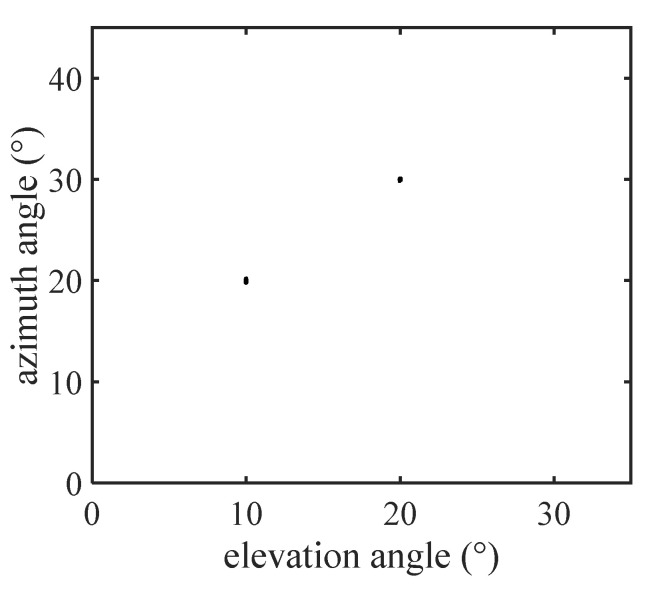
The scatter image of DOAs with TA-MUSIC.

**Figure 6 sensors-22-00136-f006:**
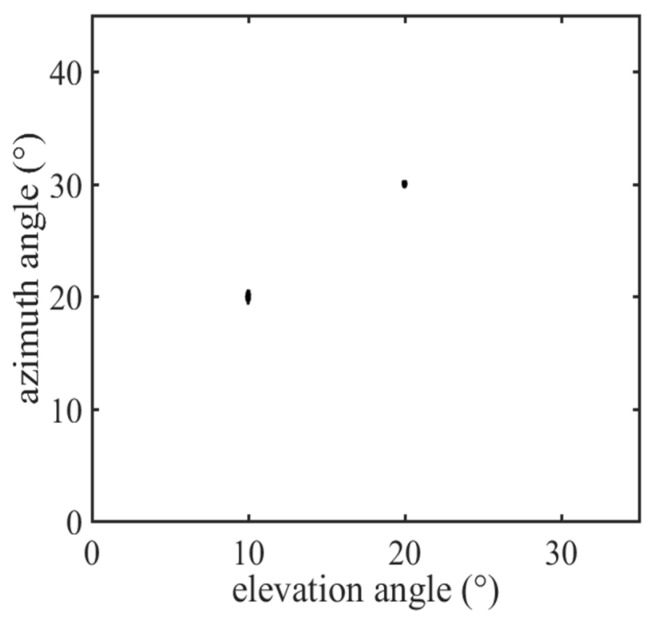
The scatter image of DOAs with E-MUSIC.

**Figure 7 sensors-22-00136-f007:**
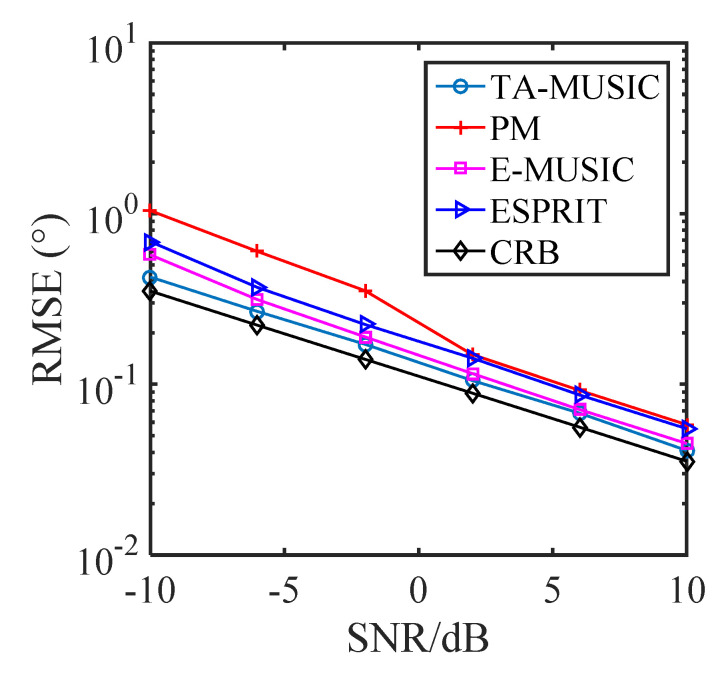
RMSE against SNR levels for different algorithms.

**Figure 8 sensors-22-00136-f008:**
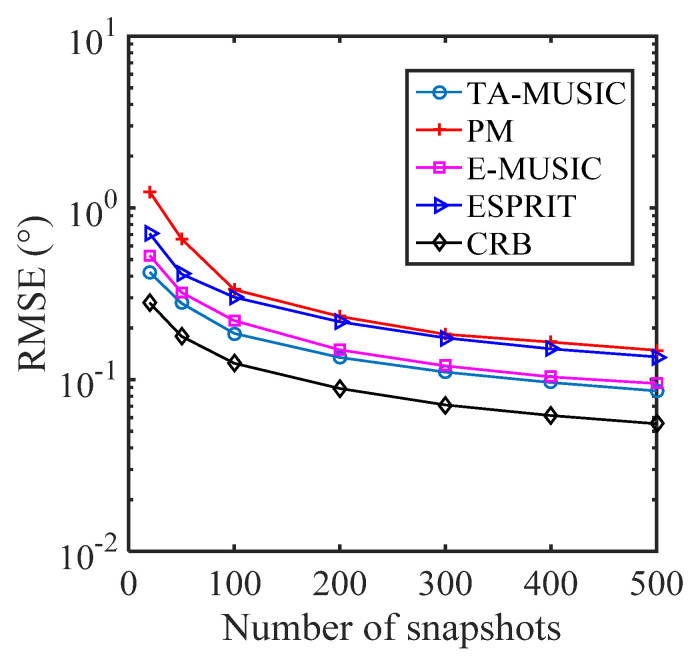
RMSE against number of snapshots for various algorithms.

**Figure 9 sensors-22-00136-f009:**
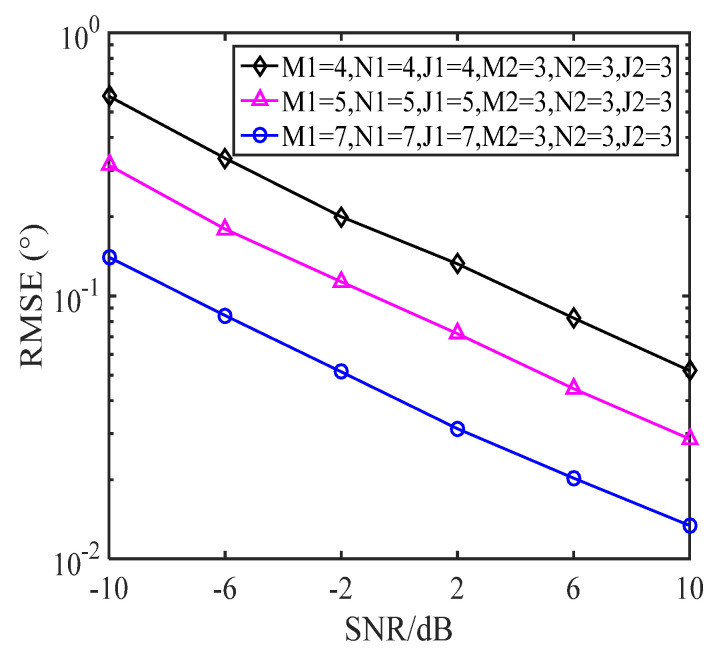
RMSE of various geometries against SNR levels.

**Figure 10 sensors-22-00136-f010:**
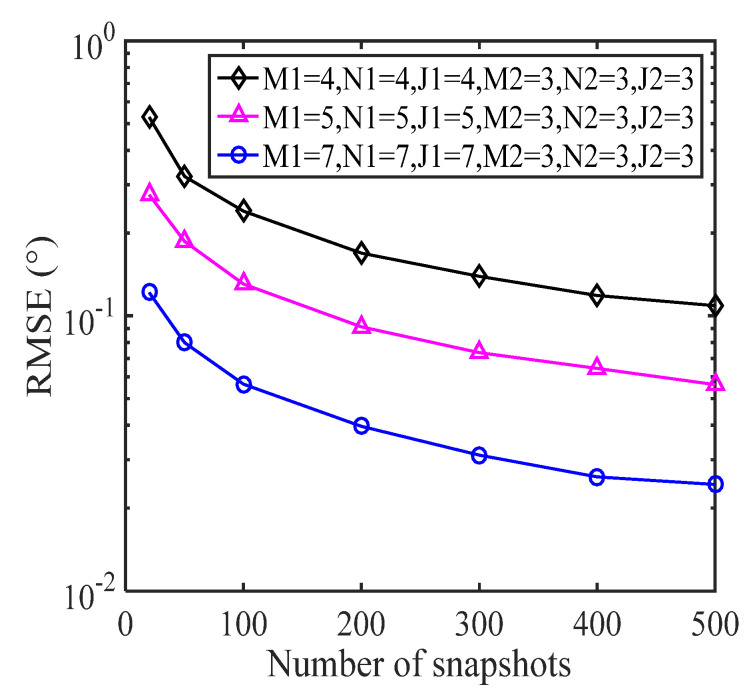
RMSE of various geometries against the number of snapshots.

**Figure 11 sensors-22-00136-f011:**
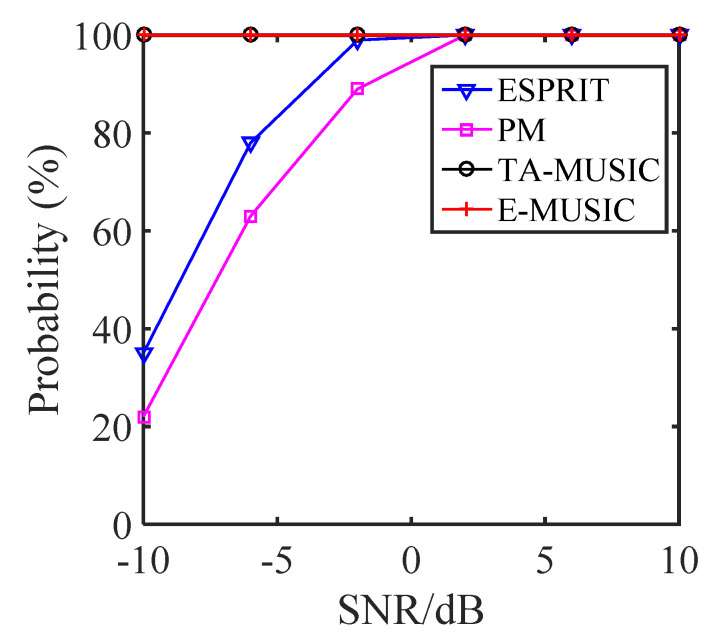
Resolution against SNR for various algorithms.

**Figure 12 sensors-22-00136-f012:**
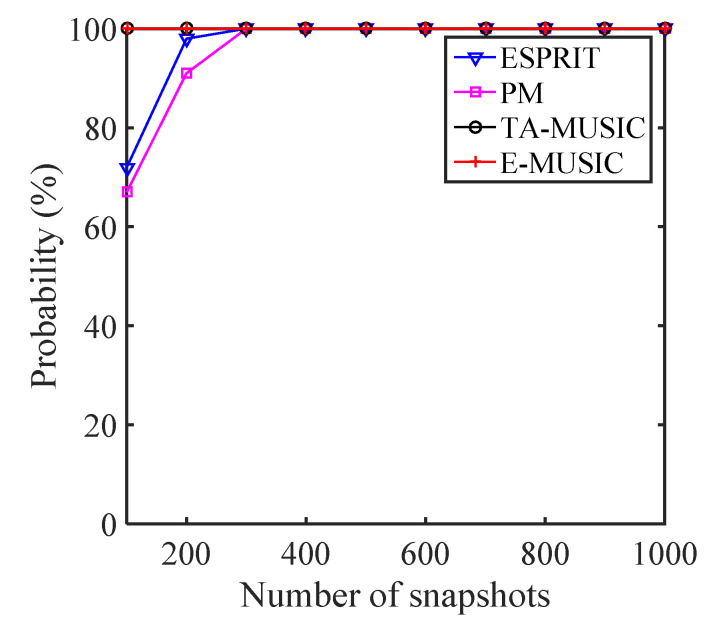
Resolution against number of snapshots for various algorithms.

**Table 1 sensors-22-00136-t001:** Computational complexity of different algorithms.

Algorithm	Complex Multiplication	Running time
Classic MUSIC	$1.880416\times 10^{12}$	3406.7221
TA-MUSIC	$3.280095\times 10^{12}$	7013.1121
E-MUSIC	$2.45380\times 10^{6}$	0.00697637

## Data Availability

Not applicable.

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
