# Peer review of "Computationally Efficient Direction-of-Arrival Estimation Algorithms for a Cubic Coprime Array"

_sensors, 2021, doi:10.3390/s22010136_

Round 1

Reviewer 1 Report

Please, find in attached file my comments and suggestions.

Reviewer 2 Report

The paper presents the problem of direction of arrival (DOA) estimation for massive multi-input multi-output (MIMO) radar, and propose a total array based multiple signals classification (TA-MUSIC) algorithm for two-dimensional direction of arrival (DOA) estimation with coprime cubic array (CCA). 

Before being suitable for publication, the authors should address these minor comments:

  1. Please rework the X and Y axes of Figure 2.
  2. A comparison table between the proposed method and the previous works is welcome to complete the introduction section and highlight the novelties of the proposed method.
  3. A more detailed description and presentation of each of the result graphs would be appreciated

Round 2

Reviewer 1 Report

The authors revised the manuscript carefully, I'm generally satisfied with the revisions and improvements.